# Replication Study of DECAF: Generating Fair Synthetic Data Using Causally-Aware Generative Networks

# 1 Summary

## 1.1 Scope of reproducibility

In this paper we attempt to reproduce the results found in "DECAF: Generating Fair Synthetic Data Using Causally-Aware Generative Networks" by Breugel et al [2]. The goal of the original paper is create a model that intakes a biased dataset and outputs a debiased synthetic dataset that can be used to train downstream models to make unbiased predictions both on synthetic and real data.

## 1.2 Methodology

We built upon the (incomplete) code provided by the authors to repeat the first experiment of [2] which involves removing existing bias from real data with existing bias, and the second experiment where synthetically injected bias is added to real data and then removed.

## 1.3 Results

We reproduced most of the data utility results reported in the first experiment for the Adult dataset. However, the fairness metric generally match the original paper but are numerically not comparable in absolute or relative terms. For the second experiment, we were unsuccessful in reproducing results found by the authors. We note however that we made considerable changes to the experimental setup, which may make it difficult to perform a direct comparison of the results.

## 1.4 What was easy

The smaller size and tabular format of both datasets allowed for quick training and model modifications.

## 1.5 What was difficult

There are several possible interpretations of the paper on both a methodological and conceptual level. Reproducing the experiments required rewriting or adding large sections of code. Given these multiple interpretations it was difficult to be confident in the reproduction. In addition, several results found by the authors appear to be counterintuitive, such as algorithms debiasing without being designed to do so and sometimes outperforming debiasing algorithms on the same dataset.

## 1.6 Communication with original authors

We sent two emails to the authors describing our issues. We received a reply with a few extra files, but no direct answer to content questions.

# 2 Introduction

It is broadly acknowledged that real world data contains bias. Despite efforts to make data collection more equitable and representative, a myriad of challenges remain. The effects of bias are well understood, as biased data can lead to the under-representation of particular demographics, such as the case of political representation in the United States Census[7]. As technology progressed to the emergence of machine learning (ML) models, the same challenges persisted as ML models adopted the biases of the data and humans who created them. Models trained on biased data can pass bias downstream to various other applications, a phenomenon referred to as algorithmic bias[5]. Such models have potential to not only perpetuate but exacerbate social inequality, yet bias is omnipresent in everything that humans touch. Hence, there is a clear and present need for methods that can utilize biased data to produce unbiased results.

# 3 Background

The notion of using Generative Adversarial Networks (GAN) to increase fairness within artificial intelligence is broadly supported by the literature. Various models exists such as FairGAN[10], GANSAN[1], and Fairness GAN [9] to name but a few. Notably, fairness efforts have typically recognized a fairness-accuracy trade-off assumption, where a fairer algorithm comes at the cost of accuracy. However, recent work has challenged these assumptions, finding that the accuracy cost of fairness is negligible in some circumstances[8]. Nonetheless, given the increased awareness of the nefarious effects of data bias, many research efforts have been directed towards the debiasing of data and other attempts to create fairer artificial intelligence.

## 3.1 DECAF premise

One such effort and the subject of the present study is DEbiasing CAusal Fairness (DECAF) [2]. DECAF takes a distinct approach to debiasing data, explicitly approaching fairness from a causal standpoint with a goal of downstream model fairness. There are three broad approaches to fairness that may be identified, (1) the preprocessing approach, where the characteristics of the input data are changed to suppress undesirable biases [2], (2) the algorithmic modification approach, where the learning algorithm itself is adapted to reduce bias [4], and (3) the postprocessing approach, where the output of a model is manipulated to obtain the desired level of fairness[6]. The DECAF approach falls in the first category of preprocessing because it attempts to remove bias from the input data and subsequently from all downstream models.

The DECAF model is a generative adversarial network (GAN) that utilizes the causal structure of directed acyclical graphs (DAGs) to remove bias from real data. The three critical assumptions of the DECAF method are (1) the data generating process is represented by a DAG, (2) the DAG is causally sufficient, and (3) the DAG is known for a given dataset. DAGs are central to the method, as it is through edge manipulation that debiasing is performed.

The model may be separated into two stages. During the first training phase, the model learns the causal conditionals of the dataset from its DAG. In the second inference phase, the data is debiased through DAG modification. Each fairness level defines a unique set of edge removals from the original DAG, resulting in a new, intervened DAG. These intervened DAGs are given to the model to generate synthetic, fair datasets from the original data. The synthetic datasets have similar distributions to the original data, but avoid bias. Because the method debiases at inference time, retraining the model is not required when using different fairness measures, thus providing inference-time fairness.

Once DECAF generates a synthetic and unbiased dataset, a simple multilayer perceptron (MLP) is trained on this synthetic data to create an unbiased classifier that can be used both on the original data and in other settings. Because the data used for training the MLP has already been debiased, the authors claim that the MLP or any chosen downstream model is guaranteed to be fair since it doesn't incorporate any of the bias from the original training data; this is a hallmark of the preprocessing approach to fairness.

## 3.2 Fairness standards

Three definitions of algorithmic fairness are used in the paper, each corresponding to a unique modified DAG. The most lenient standard is the commonly used Fairness Through Unawareness (FTU) definition, which entails that the protected variable, $A$, is not explicitly used by the model to predict the label, $\hat{Y}$. While widely used because it avoids direct discrimination, FTU fails to eliminate indirect discrimination.

A more stringent definition of fairness is Demographic Parity (DP), which declares that classification probability must be independent of classes, i.e. if the protected attribute is gender, all gender classes have the same success rate. The DP definition is considered to be very strict because it potentially under-utilizes feature differences between groups in the process of blocking indirect discrimination.

Conditional Fairness (CF) lies in the middle ground between the first two definitions by presuming that the selection rate between groups segregated by the protected attribute must be the same when conditioned on some explanatory variable(s) determined by prior knowledge. Each of these standards corresponds to a variation of DECAF, respectively DECAF-ND (no debiasing), DECAF-FTU, DECAF-CF, and DECAF-DP. The fairness of each model is tested against FTU and DP metrics.

## 4    Scope of reproducibility and claims

The authors claim that DECAF allows for the generation of unbiased synthetic data from biased real data and that their method does so with minimal loss in data utility compared to other approaches. Furthermore, they identify five characteristics of fair synthetic data that their method achieves: (1) allows post-hoc distribution changes, (2) provides fairness, (3) supports causal notions of fairness, (4) allows inference-time fairness, and (5) requires minimal assumptions. Additionally, they claim that DECAF is the only method to achieve all of the five listed characteristics.

The authors identify three main contributions of their work:

(i) DECAF, a causal GAN-based model that can use a biased dataset $X$ to generate an equivalent synthetic unbiased dataset $\mathcal{X}$ with minimal loss of data utility

(ii) A flexible causal approach for modifying DECAF to generate fair data

(iii) Guarantee that downstream models trained on the generated synthetic data will make unbiased predictions on both synthetic and real-life (biased) data

We aim to evaluate claims (i) and (iii) by replicating the two experiments of [2]. We will focus on the narrow interpretation of reproducibility, namely whether the experiment can be reproduced by independent researchers with the same setup rather than testing against the more general standard of replicatability on different datasets. Despite the availability of code, there were considerable problems with running the models even with instructions given, meaning that we limited our scope to direct reproducibility. As the authors have done, we will evaluate the data utility of the DECAF method with precision, recall, and area under the receiver operation characteristic (AUROC); fairness will be evaluated with Fairness Through Unawareness (FTU) and Demographic Parity (DP) measures.

## 5    Methodology

While code from the creators of the DECAF method is available [1], documentation leaves room for interpretation and the instructions given for running the code do not reproduce the results as presented. In addition, there are several possible discrepancies between the method described in the paper and the code provided. Thus, we made the assumption that the paper leads and adjusted the code accordingly to match.

### 5.1    Methodological Code Changes

Though the DECAF class was working, several components of the experimental setup code was either missing or not fully explained. Thus, we had to extrapolate heavily to produce results. The major code changes required are listed below:

---

[1]The DECAF code is available at: https://github.com/vanderschaarlab/DECAF

(i) Preprocessing: the paper mentioned standardizing continuous variables, however, following the procedure given in the paper generated uninterpretable results. As a solution we attempted to standardize all variables, including categorical ones though we question the conceptual validity of this decision. After standardizing with StandardScaler, we still were not getting results as high as the reported metrics, so we tried normalizing with MinMaxScaler which finally produced matching results in data utility. The DECAF class employs a final sigmoid layer that converts all generated data to a range between 0 and 1. We suspect this was the reason why their `run_example.py` script would only predict labels of one class and why using a Scaler allowed us to obtain meaningful predictions.

(ii) DAGs: There appears to be a mismatch with the dags provided, as neither contain all of the variables in the datasets. In addition the code provided utilized a toy graph. The authors state that they used `Tetrad` to generate the DAG for the dataset, so we attempted to generate a full causal graph for the Adult dataset, but our generated graphs did not match Figure 6 and 7 of [2]. Hence, we manually input the graphs from the paper.

(iii) Label Generation: The paper instructed that the labels for synthetic data should be generated by the model as they are part of the causal dependencies graph. The original code did not generate the labels for the synthetic dataset, but instead generated only the x values and then predicted the labels from those generated x values using the baseline model. The code seemed to omit the target variable from the GAN input, but we felt this would leave out valuable causal information contained in the edges from the explanatory variables to the target variable. Thus, we decided to include the target variable in the DAG, and this indeed improved our results. In the end, we were forced to generate labels for experiment 1, while predicting labels for experiment 2 in order to obtain interpretable results.

(iv) Downstream Classifer: The paper mentions an MLP from `sklearn`, but the example code uses an XGBClassifier as the downstream classifier which was giving us installation issues. We followed the paper by using an MLP.

## 5.2 Dataset

For the first experiment, we worked with the Adult dataset [2] [3] collected from the 1994 United States Census. The dataset contains about 45,000 data points, and 2,000 data points were set aside for the test set as specified by [2]. The protected attribute is sex, and the target variable is income with roughly 75% in the '<=50k' class and the remaining 25% belonging to the '>50k' class. This makes sense considering the average earnings of Americans at the time, but does make our data rather skewed towards one class. We manually input the DAG from Figure 6 of [2] and used the preprocessing steps described in the previous section.

For the second experiment, we used the Credit Approval dataset [3] of credit card applications. This dataset is considerably smaller than the first dataset with only 678 data points. The original paper did not specify how large the test set was, so we chose a typical 80%/20% split for training and testing. The protected attribute is ethnicity and the target variable is application approval. About 55% of the applications were approved while the rest were rejected, so this dataset is considerably more balanced than the other. Again, we had to manually input the graph from Figure 7 of the original paper. Since the protected attribute here, ethnicity, is not binary, we first converted the variable to be binary with 0 corresponding to 'not discriminated against' and 1 to 'discriminated against'. Then we used the same preprocessing steps as in the first experiment.

## 5.3 Hyperparameters

A hyperparameter search is not necessary for our experiments. We used the DECAF class as given with the parameters set by the authors' code. The only modification we made was changing the `dag_seed` parameter from the provided toy graph to the respective graphs for each dataset presented on Page 28 of [2]. The DECAF generator is instantiated with $d$, the number of features, sub-networks with shared hidden layers. The generator and discriminator both use 2 hidden layers with $2d$ neurons. The generator is updated once for every 10 discriminator updates. Adam was used as the optimizer with a learning rate of 0.001. The other GANs used for comparison were also given default parameters and settings from their respective packages because no settings were specified by the authors.

---

[2]The Adult dataset is available at `http://archive.ics.uci.edu/ml/index.php`

Table 1: Reproduction results on bias removal experiment on the Adult dataset.

| Method | Data Quality | | | Fairness | |
|---|---|---|---|---|---|
| | Precision | Recall | AUROC | FTU | DP |
| Original data | **0.881**±0.006 | $0.917 \pm 0.009$ | **0.772**±0.008 | $0.047 \pm 0.010$ | $0.207 \pm 0.013$ |
| GAN | $0.772 \pm 0.098$ | $0.344 \pm 0.249$ | $0.523 \pm 0.048$ | $0.202 \pm 0.197$ | $0.202 \pm 0.182$ |
| WGAN-GP | $0.784 \pm 0.073$ | $0.467 \pm 0.195$ | $0.514 \pm 0.067$ | $0.208 \pm 0.189$ | $0.231 \pm 0.166$ |
| FairGAN | $0.835 \pm 0.043$ | $0.911 \pm 0.081$ | $0.672 \pm 0.061$ | $0.097 \pm 0.113$ | $0.157 \pm 0.155$ |
| DECAF-ND | $0.880 \pm 0.024$ | $0.774 \pm 0.047$ | $0.734 \pm 0.023$ | $0.114 \pm 0.040$ | $0.353 \pm 0.023$ |
| DECAF-FTU | $0.866 \pm 0.027$ | $0.800 \pm 0.043$ | $0.708 \pm 0.043$ | $0.041 \pm 0.020$ | $0.260 \pm 0.085$ |
| DECAF-CF | $0.769 \pm 0.012$ | $0.954 \pm 0.025$ | $0.541 \pm 0.028$ | $0.022 \pm 0.018$ | $0.026 \pm 0.023$ |
| DECAF-DP | $0.753 \pm 0.003$ | **0.978**±0.022 | $0.502 \pm 0.009$ | **0.006**±0.007 | **0.012**±0.009 |

An MLP with default parameters from `sklearn` was used. The default settings are 100 neurons with ReLU activation functions and Adam with a learning rate of 0.001. A Softmax activation and binary cross entropy loss were used for the output layer.

## 5.4 Experimental setup and code

In this study, we aimed to replicate the experiments of the original paper, Debiasing Census Data (experiment 1) and Fair Credit Approval (experiment 2), to evaluate the performance of DECAF when generating unbiased synthetic data from real, biased data from the Adult dataset.

We trained each model listed in Table 2 of the original paper, four DECAF GANs and three other GANs for comparison, for 50 epochs. A synthetic dataset was generated from each model that was then used to train an MLP to classify a test set of 2,000 unmodified data points from the original dataset. We compared these predictions with the ground truth labels from the original data to evaluate performance and fairness. This process was repeated ten times to obtain average metrics over multiple runs as specified by the authors.

To mimic the DECAF paper, precision, recall, and AUROC were used to measure the performance of the models, while FTU and DP were used to measure the fairness of the models. Precision, recall, and AUROC are given by `sklearn.metrics`, and higher scores indicate better performance. Lower FTU and DP scores indicate less bias. To calculate FTU, set all the labels of the protected attribute to one class and predict the labels; repeat with the remaining class (for binary variables), and compare the difference of the means of the two prediction sets, such that $|P_{A=0}(\hat{Y}|X) - P_{A=1}(\hat{Y}|X)|$ Then for DP, segregate the dataset into datapoints with one class label and datapoints with the other label (for binary variables), and again predict the labels of each set and compare the difference of the means of the two prediction sets, such that $|P(\hat{Y}|A = 0) - P(\hat{Y}|A = 1)|$. To compare our replication against the original experiments of the authors, we compare both the absolute difference and the relative difference (as a ratio) with our findings. Our code and more details can be found on our Github repository[3].

## 5.5 Computational requirements

Because the datasets used are small and tabular, the computational requirements are minimal. No GPU was necessary; all models were run on an Intel Core i7-8750h CPU. It takes six minutes to train DECAF models on the Adult dataset [3] for 50 epochs, and five seconds to generate synthetic data. The total runtime is about four hours for experiment 1 and about two hours for experiment 2.

## 6 Results

We were able to reproduce some results in experiment 1, but we could not get similar results on the second experiment. Table 1 shows our result that synthetic data is generated using each benchmark method, after which a separate MLP is trained on each dataset for computing the metrics, and Table.2 is the result from the original paper. Section 5.4 details how we obtained the relevant metrics. We can see DECAF does have the effect of debiasing and there is improvement comparable with FairGAN.

---

[3]Our Github repository: `https://anonymous.4open.science/r/DECAF-CF0A/`

Table 2: Original results of bias removal experiment on the Adult dataset.

| Method | Data quality | | | Fairness | |
|---|---|---|---|---|---|
| | Precision | Recall | AUROC | FTU | DP |
| Original data | **0.920**±0.006 | **0.936**±0.008 | **0.807**±0.004 | 0.116 ± 0.028 | 0.180 ± 0.010 |
| GAN | 0.607 ± 0.080 | 0.439 ± 0.037 | 0.567 ± 0.132 | 0.023 ± 0.010 | 0.089 ± 0.008 |
| WGAN-GP | 0.683 ± 0.015 | 0.914 ± 0.005 | 0.798 ± 0.009 | 0.120 ± 0.014 | 0.189 ± 0.024 |
| FairGAN | 0.681 ± 0.023 | 0.814 ± 0.079 | 0.766 ± 0.029 | 0.009 ± 0.002 | 0.097 ± 0.018 |
| DECAF-ND | 0.780 ± 0.023 | 0.920 ± 0.045 | 0.781 ± 0.007 | 0.152 ± 0.013 | 0.198 ± 0.013 |
| DECAF-FTU | 0.763 ± 0.033 | 0.925 ± 0.040 | 0.765 ± 0.010 | 0.004 ± 0.004 | 0.054 ± 0.005 |
| DECAF-CF | 0.743 ± 0.022 | 0.875 ± 0.038 | 0.769 ± 0.004 | 0.003 ± 0.006 | 0.039 ± 0.011 |
| DECAF-DP | 0.781 ± 0.018 | 0.881 ± 0.050 | 0.672 ± 0.014 | **0.001**±0.001 | **0.001**±0.001 |

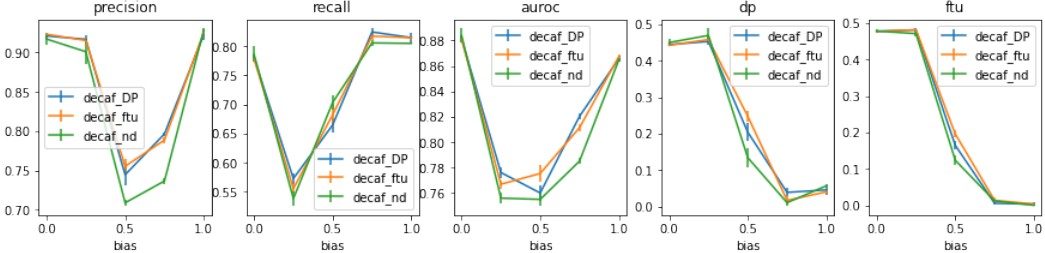

Figure 1: Plot of precision, recall, AUROC, FTU, and DP over bias strength.

Also same as in the original paper, DECAF-ND performs almost the best among all methods in terms of data quality. Methods DECAF-FTU, DECAF-CF, and DECAF-DP have relatively lower scores on data quality but perform better on fairness.

Figure 1 shows DECAF results for experiment 2 in which removing synthetically injected bias. These results do not match the Figure 3 of original paper. This mismatch is not surprising because the second experiment is based on the first experiment where we suspect our setup already significantly diverges from that of the authors.

# 7 Discussion

Overall, we have been able to produce the results found by the authors. That being said, there are multiple interpretations of the results and overall saliency is relatively low. For the purpose of this paper, we will focus primarily on the fairness metrics since the data utility metrics are closer to the findings of the authors and fairness is the primary goal of the method. Though the order of the fairness of various models of our results match with the original results from the paper, our numerical figures do not match the authors' results with a satisfactory level of precision. Several observations are further pursued as plausible explanations for this phenomenon.

## 7.1 Interpretation of the results

As shown in Tables 1 and 2, we obtained interpretable results for all models tested in experiment 1. For the most part, we found effects similar to the authors, but they deviate significantly in numerical terms. More specifically, we do find that as the model variations move from least strict to most strict definition of fairness, the fairness increases and data utility decreases. However, there are notable deviations from the authors results, specifically concerning the fairness metrics of the GAN. In addition, we find that DECAF-ND increases the level of bias compared to the original dataset which matches the authors. However, we find a higher DP of 0.353 and a FTU of 0.114 compared to the authors DP of 0.198 and FTU of 0.152. These results run counter to our expectations.

The results found in the Credit dataset also show the directional correctness of DECAF in reducing bias, but direct comparison to the authors findings is difficult because our results differ significantly from the authors' findings. In particular, we find the FTU and DP scores is maximized at, 0 and minimized at 1. In addition, the authors find relatively stable data utility metrics, whereas we find a

significant decrease between bias 0.25 and 0.75. The results for bias 1.0 and 0 do reflect the average value found by the authors, with the exception of recall which is significantly lower.

Furthermore, the authors did not directly interpret their chosen metrics. The original paper designated FTU and DP measures for fairness and reported figures, but did not explain the actual meaning of the numbers and magnitude of changes seen. For example, most of the reported fairness metrics were very small, but we did not have any guidance on the significance of a .001 decrease in the FTU metric. Thus, we felt the paper lacked explainability. Additionally, the fairness definitions themselves, the instructions for calculating the fairness measures, and the given FTU and DP code were somewhat contradictory. Calculating FTU and DP based on our interpretation of the authors' method did not reproduce their results. Using the FTU and DP calculations from an extra code file we received still did not produce matching results. One possibility is that the authors' final fairness metrics calculation code was not contained in the files we had access to and does not match any of the implementations we attempted.

## 7.2   What was easy

One aspect that eased our investigation into the reproduceability of [2] was the tabular format and small size of the datasets we used. Training and modifying the model was not computationally expensive or time consuming, thus we could test many different strategies to find the closest solution.

## 7.3   What was difficult

We were originally under the impression that the DECAF code repository was fully functional as a basis for extension. Upon further examination, we found that it was not working and did not reproduce the published results. Thus, we had to pivot from extending their code to replicating the results with our own code which was challenging in itself. While attempting to reproduce the experiments, we found that the instructions given were incomplete and contradictory to the code provided.

There are multiple obstacles to replicating the experiments as described, which can broadly be separated into conceptual and methodological issues. On the former, there are many important research decisions that are not fully articulated, as well as results that appear counterintuitive. For example, the authors found that their application of GAN, a method that does not do explicit debiasing, had significantly improved fairness metrics compared to the original dataset. One would expect that all the methods that do not debias, namely original data, GAN, WGAN-GP and DECAF-ND would perform in the same order of magnitude in terms of fairness, but this is not the case in the author's initial findings. Moreover, while the DECAF models do reduce bias in line with the level of fairness required, DECAF-ND actually makes the dataset more biased compared to the original dataset. Our reproduction of GAN does match the expected results, with original data, GAN, and WGAN all returning roughly the same fairness metrics. As discussed, we successfully reproduced the overall impact of DECAF, namely higher fairness and lower data utility for more stringent definitions of fairness. However, DECAF-ND exhibits considerably higher bias than the original dataset and no clear intuition is given on why this may be the case.

In addition to the conceptual challenges, there are multiple methodological issues. Following the instructions provided by the authors resulted in numerous compatibility warnings and failed tests. As described in section 5.1, several substantial changes were needed to generate any interpretable results. Further compounding these issues, there are inconsistencies in the applied method, as the code utilized in the example explicitly deviates from the approach described in the experimental setup. We were forced to generate labels for experiment 1, while predicting labels for experiment 2. Attempts to use generated labels made experiment 2 uninterpretable, as all key performance indicators would become zero otherwise. This methodological inconsistency between experiments further problematizes the reproducibility of DECAF.

## 7.4   Overall reproducibilty

Due to the number of possible conceptual and methodological interpretations with the code, modifications were needed as described in section 5.1. While we were successful in producing results that could be interpreted, the numerical variations and methodological deviations are so substantial

that further research would be needed to assess the overall accuracy of the authors claims. We found evidence that supports the narrow interpretation of the claims made by the author, namely that DECAF reduces bias in downstream models, and allows for the generation of debiased synthetic data. However, the authors claim that the approach allows for minimal data utility loss. Without a further explanation on what is considered minimal data utility loss, it is difficult to evaluate this claim, especially with amount of deviation found between the authors results and ours. While our findings on the first experiment are in line with the authors, the results of the second experiment are in direct contradiction to their findings. Since any fundamental issues in experiment 1 are likely to carry over to experiment 2 we focus our recommendations on experiment 1.

Overall, we find that the results are reproducible but difficult to interpret and compare. Fruitful avenues of further investigation would be to re-evaluate the fairness metrics. Another hypothesis is that there is a more functional issue with the DECAF model itself that would lend itself to further investigation.

### 7.5 Communication with original authors

We sent two emails to the authors of DECAF detailing the aforementioned code issues. One author did respond with a few extra code files, but unfortunately did not directly address out content questions. However, several of the interpretations we made were retroactively confirmed by the extra code files.

## 8 Conclusion

During our investigation, we faced multiple significant challenges in reproducing the results of the original paper. The biggest challenges stemmed from the number of possible interpretations of the code and method. While we were not able to reproduce the results in full, we believe methods like DECAF have great potential for expansion. The relevance of unbiased downstream classifiers and the evident need for bias removal in real data will likely remain a societally relevant area of research. For instance, the Adult dataset[3] we studied is nearing 30 years old. Perhaps an intriguing next phase could be to pull this year's Census data to investigate how bias has changed over time and if DECAF is still applicable for removing likely more nuanced and hidden bias that persists through the increased awareness of bias and techniques for counteracting bias that exist today.

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

343 8622525.

# 9 Appendices

Table 3: Absolute difference between authors' findings and our results.

| Method | Data quality | | | Fairness | |
|---|---|---|---|---|---|
| | Precision | Recall | AUROC | FTU | DP |
| Original data | 0.109 | 0.046 | .807 | 0.116 | .180 |
| GAN | −0.165 | 0.095 | 0.044 | −0.179 | −0.113 |
| WGAN-GP | −0.101 | 0.447 | 0.284 | −0.088 | −0.042 |
| FairGAN | −0.154 | −0.097 | 0.094 | −0.088 | −0.06 |
| DECAF-ND | −0.107 | 0.143 | 0.047 | 0.038 | −0.155 |
| DECAF-FTU | −0.103 | 0.125 | 0.057 | −0.037 | −0.206 |
| DECAF-CF | −0.026 | −0.079 | 0.228 | −0.019 | 0.013 |
| DECAF-DP | 0.028 | −0.097 | 0.17 | −0.005 | −0.011 |

345 Absolute difference is calculated as the value found by the authors minus the value found in our
346 reproduction.

Table 4: Performance relative to original data from authors.

| Method | Data quality | | | Fairness | |
|---|---|---|---|---|---|
| | Precision | Recall | AUROC | FTU | DP |
| Original data | 1 | 1 | | 1 | |
| GAN | 0.66 | 0.46 | 0.70 | 0.20 | 0.49 |
| WGAN-GP | 0.74 | 0.95 | 0.98 | 1.03 | 1.05 |
| FairGAN | 0.74 | 0.85 | 0.95 | 0.08 | 0.54 |
| DECAF-ND | 0.85 | 0.96 | 0.97 | 1.31 | 1.10 |
| DECAF-FTU | 0.83 | 0.96 | 0.95 | 0.03 | 0.30 |
| DECAF-CF | 0.81 | 0.91 | 0.95 | 0.3 | 0.22 |
| DECAF-DP | 0.85 | 0.91 | 0.83 | 0.01 | 0.01 |

347 Relative performance is calculated as the ratio between the original data and the performance of the
348 selected model on the same variable.

Table 5: Performance relative to original data in our findings.

| Method | Data quality | | | Fairness | |
|---|---|---|---|---|---|
| | Precision | Recall | AUROC | FTU | DP |
| Original data | 1 | 1 | | 1 | |
| GAN | 0.95 | 0.38 | 0.72 | 4.30 | 0.98 |
| WGAN-GP | 0.97 | 0.51 | 0.71 | 4.43 | 1.12 |
| FairGAN | 1.03 | 0.99 | 0.93 | 2.06 | 0.76 |
| DECAF-ND | 1.09 | 0.85 | 1.02 | 2.43 | 1.70 |
| DECAF-FTU | 1.07 | 0.87 | 0.98 | 0.87 | 1.26 |
| DECAF-CF | 0.95 | 0.104 | 0.75 | 0.47 | 0.13 |
| DECAF-DP | 0.93 | 1.07 | 0.70 | 0.13 | 0.06 |

Table 6: Reproduction results on bias removal experiment on the Credit dataset.

| Method | Data quality | | | Fairness | |
|---|---|---|---|---|---|
| | Precision | Recall | AUROC | FTU | DP |
| Original data | **0.915**±0.007 | $0.787 \pm 0.009$ | **0.840**±0.004 | **0.013**±0.008 | **0.011**±0.007 |
| DECAF-ND | $0.809 \pm 0.083$ | $0.813 \pm 0.047$ | $0.758 \pm 0.080$ | $0.085 \pm 0.035$ | $0.053 \pm 0.035$ |
| DECAF-FTU | $0.821 \pm 0.072$ | $0.811 \pm 0.050$ | $0.770 \pm 0.055$ | $0.032 \pm 0.028$ | $0.065 \pm 0.040$ |
| DECAF-DP | $0.784 \pm 0.064$ | **0.836**±0.047 | $0.744 \pm 0.055$ | $0.045 \pm 0.036$ | $0.063 \pm 0.030$ |

