# OpenReview forum: "Replication Study of DECAF: Generating Fair Synthetic Data Using Causally-Aware Generative Networks"
_ML_Reproducibility_Challenge/2021/Fall — RC2021_

### Official Review · Reviewer_NrN7 · 2022-03-01

**Rating:** 7
**Confidence:** 4

**Review:**

Overall, I think this is a nice replication of the original paper. It is unfortunate that the original released code was not adequate for a simple reproduce, but commend the authors for their tenacity in their efforts. Some suggestions, but overall I think the replication should be accepted.

Suggestions:
- I don't think you mention how the error bars in table 1 (reproduction) and 2 (original) are generated. You should report what they are here even if they are reported in the original paper.
- It would have been interesting to use this method with a different dataset to try and test the robustness and generalizability of the proposed methods. While this isn't necessary for acceptance because of the challenges you faced in reproduction, maybe would be a nice contribution for the final report.
- There are a few places that are a bit difficult to read and should be edited/rewritten to be made clearer. Also some typos are included in this list.
   - Line 193: missing period after the equation
   - Line 235: This paragraph is a bit hard to parse for me. You should also try to be consistent about 1 vs 1.0
   - Line 305: "out content questions" -> "our content questions"
   - I think there are a few other places, so you should just go through with a fined tooth comb a few more times.

---

### Official Review · Reviewer_U2zT · 2022-03-20

**Rating:** 7
**Confidence:** 4

**Review:**

This paper describes the reproducibility process and results for the Decaf paper.

Overall, the authors have conducted thorough experimentation for the paper and analyzed the results accordingly. Here are my comments on the paper.

1.  Some of the statements discussed in the introduction might be a little strong. For example, the authors said "The effects of bias are well understood," Im not sure if this is entire true, although it is generally agreed on that bias in the data can impact learning, I'm not sure if the effects of the bias are fully understood. It would be nice if the authors could modify these statements.

2. The authors have conducted thorough experimentation in attempt to reproduce the results in the Decaf paper. The authors described clearly the challenges they faced while attempting the reproduce the experimental results. the authors have also conducted carefully analysis of the reproduction results.

3. The methodology, experiments and discussions are well written and easy to follow.

Overall, I think the authors have conducted thorough experiments in order to reproduce the results. The paper is well written and easy to follow.

---

### Meta-Review · Program_Chairs · 2022-04-07

**Recommendation:** Accept
**Confidence:** 5

**Metareview:**

A good-quality report with some interesting findings regarding the reproducilibity of the DECAF paper. There are some minor points that were identified by reviewers regarding result presentation and the way in which things are communicated (i.e. when you make strong statements like "The effects of bias are well understood", they should be accompanied by a citation proving this point).

---

### Decision · Program_Chairs · 2022-04-09

**Decision:**

Accept

**Comment:**

Following the recommendation of reviewers and meta-reviewer, the paper is accepted for ML Reproducibility Challenge 2021, and will be published in the upcoming special edition of ReScience Journal.